# *LNX1* Contributes to Cell Cycle Progression and Cisplatin Resistance

**DOI:** 10.3390/cancers13164066

**Published:** 2021-08-12

**Authors:** Minsu Jang, Rackhyun Park, Yea-In Park, Yeonjeong Park, Jin I. Lee, Sim Namkoong, Eun-Ju Lee, Junsoo Park

**Affiliations:** 1Division of Biological Science and Technology, Yonsei University, Wonju 26493, Korea; minsujang@yonsei.ac.kr (M.J.); rockhyun@yonsei.ac.kr (R.P.); pyi012324@yonsei.ac.kr (Y.-I.P.); bbling408@yonsei.ac.kr (Y.P.); jinillee@yonsei.ac.kr (J.I.L.); 2Department of Biochemistry, Kangwon National University, Chuncheon 24341, Korea; simn@kangwon.ac.kr; 3Department of Obstetrics and Gynecology, School of Medicine, Chung-Ang University, Seoul 06973, Korea; ejlee@cau.ac.kr

**Keywords:** *LNX1*, cell cycle, cisplatin, cell death

## Abstract

**Simple Summary:**

The ligand of numb-protein X1 (*LNX1)* is reported to be upregulated in various cancers, however the cellular function of *LNX1* is not clearly characterized. The aim of the present study was to elucidate the regulation of *LNX1* expression and clarify the role of *LNX1* in cell-cycle progression and resistance to the cancer therapeutic agent, cisplatin. We found that LNX1 expression is decreased by DNA damage including cisplatin treatment and the levels of S and G2/M populations were correlated with *LNX1* expression. We also showed that the upregulation of LNX1 contributes to cell-cycle progression and cisplatin resistance. Our data suggest that *LNX1* is the important regulator of the cell cycle, and contributes to tumor progression.

**Abstract:**

The ligand of numb-protein X1 (*LNX1*) acts as a proto-oncogene by inhibiting *p53* stability; however, the regulation of *LNX1* expression has not been investigated. In this study, we screened chemicals to identify factors that potentially regulate *LNX1* expression. We found that *LNX1* expression levels were decreased by DNA damage, including that by cisplatin. Upon treatment with lipopolysaccharide (LPS) and phorbol 12-myristate 13-acetate (PMA), *LNX1* expression levels increased. In addition, cell-cycle progression increased upon *LNX1* expression; the levels of S and G2/M populations were correlated with *LNX1* expression. Moreover, in CRISPR-Cas9-mediated *LNX1* knockout cells, we observed a delay in cell-cycle progression and a downregulation of genes encoding the cell-cycle markers cyclin D1 and cyclin E1. Finally, the upregulation of *LNX1*-activated cell-cycle progression and increased resistance to cisplatin-mediated cell death. Taken together, these results suggest that *LNX1* contributes to cell-cycle progression and cisplatin resistance.

## 1. Introduction

The ligand of numb-protein X1 (*LNX1*) is a binding partner for the cell-fate-determinant *NUMB* [1]. There are two isoforms of *LNX1*: *LNX1* p80 and *LNX1* p70. *LNX1* p80 has been hereafter referred to as *LNX1* [2,3]. It is a RING-type E3 ubiquitin ligase that contains a catalytic N-terminal RING domain, an amino acid motif NPAY for binding with phosphotyrosine-binding domains, and four PDZ domains. *LNX1*, through its RING domain, can ubiquitinate specific isoforms of *NUMB* and induce their proteasomal degradation [1]. A shorter, alternatively spliced isoform, *LNX1* p70, is deficient in the N-terminal RING domain and is distributed in the brain and kidney glomeruli. *LNX1* p70 interacts with cell adhesion molecules to reorganize cell junctions and acts as a scaffold to promote the ubiquitination of its ligands by recruiting other E3-ligases [4]. Different studies have predicted and identified the binding partners of *LNX1* [5,6]. *LNX1* interacts with proto-oncogenes such as erythroblastic leukemia viral oncogene homolog 2 (*ERBB2*) and *c-Src* through its PDZ domain [7,8]. Moreover, it can interact with *pJAK2* to regulate lung cancer [9]. In glioblastoma, *LNX1* regulates *Notch1* signaling and induces expansion of the glioma stem cell population [10]. Furthermore, several studies associated with cancer and *LNX1* have suggested its role as a tumor regulator in various cancers, including gliomas and colorectal carcinoma [11,12]. LNX1 related protein, LNX2 is also reported to upregulate Wnt/beta-catenin pathway in colorectal cancers [13]. In addition, recent studies also showed the role of LNX1 in the nervous system and pathophysiological cell signaling [14,15].

Previously, we reported that *LNX1* acts as a proto-oncogene by inhibiting *p53* stability [16]. *p53* is a major tumor suppressor that regulates numerous signaling pathways including those involved in cell death, growth, DNA repair, and cellular senescence [17]. In the previous study, we have shown that *LNX1* interacts with *p53* and is ubiquitinated in an *MDM2*-dependent manner. In CRISPR-Cas9-mediated *LNX1* knockout (KO) cells, tumor growth was reduced because of increased *p53* stability. In addition, we demonstrated that *LNX1* enhanced tumor growth both in cell culture and xenograft models by inhibiting a *p53*-dependent signaling pathway in *p53* wild-type (WT) cancer cells [16]. However, the regulation of *LNX1* remains unclear.

Cisplatin is one of the most widely used anticancer drugs [18]. However, its clinical efficacy is limited owing to frequently emerging cisplatin-resistant cell populations [18,19,20]. Several cellular changes have been implicated in cisplatin resistance, including increased glutathione or metallothionein content, decreased accumulation and/or increased efflux of cisplatin, and increased DNA repair [21]. Moreover, additional oncogenic pathways (such as *ras*, *c-fos/AP1*, and *bcl-2* pathways) have been characterized, which indicate that the altered expression of oncogenes could subsequently limit cisplatin-mediated DNA damage. Additionally, the activation of antiapoptotic pathways may contribute to cisplatin resistance [22,23].

In this study, we found that DNA-damage inducers decreased *LNX1* expression levels; however, LPS and PMA increased these levels. In addition, the cell-cycle progression increased *LNX1* expression levels, and S, G2, and M populations correlated with these levels. Moreover, in CRISPR-Cas9-mediated *LNX1* KOs, cell-cycle progression was delayed and the expression levels of the genes encoding the cell-cycle markers cyclin D1 and cyclin E1 decreased. Our study suggests that *LNX1* expression contributes to cell-cycle progression. Here, we investigated the potential role of *LNX1* in cisplatin resistance using overexpression experiments. We found that the overexpression of *LNX1* induces cell death upon cisplatin treatment.

## 2. Materials and Methods

### 2.1. Cell Culture and Cell Viability Assay

A549 human lung cancer cells, MCF7 human breast cancer cells, H460 human lung cancer cells and HEK293 and HEK293T human embryonic kidney cells were maintained in DMEM (WelGene, Gyeongsan-si, South Korea) supplemented with 10% fetal bovine serum (FBS; Thermo Fisher Scientific, Waltham, MA, USA) and antibiotic-antimycotic solution (WelGene). HEK293 and HEK293T cells were transfected by using Lipofectamine (Thermo Fisher Scientific), according to the manufacturer’s protocol, and calcium phosphate, respectively. Cell proliferation was measured using the 3-(4,5-dimethylthiazol-2-yl)-2,5-diphenyltetrazolium bromide (MTT) assay. Briefly, cells were seeded uniformly into the wells of a 24-well plate. At the indicated time, a final concentration of 1 mg/mL of the MTT solution was added; the mixture was incubated for an additional 3 h. MTT was purchased from Thermo Fisher Scientific. Live cells were counted using a Vi-Cell XR automatic cell counter (Beckman Coulter, Indianapolis, IN, USA), and trypan blue dye was used to differentiate between the live and dead cells. Next, the resuspended cells were collected and delivered to the flow cell for imaging purposes. The counter was set to capture 50 images per sample. Three cycles of both the aspirate and trypan blue were used to maintain single-cell suspension and achieve sufficient mixing.

### 2.2. Cell Line Generation

The generation of *LNX1* KO cell lines with CRISPR-Cas9 has been described previously [16]. For *LNX1* stable cell lines, *LNX1* CDSs were cloned into HA-pCDNA3 vectors. HEK293 cells were transfected using Lipofectamine 2000 reagent (Invitrogen, Carlsbad, CA, USA). Twenty-four hours prior to transfection, cells were seeded at a density of 1.5 × 10^5^ cells/well in 6-well culture plates. Transfection was performed at 60% confluency with 2 μg DNA and 4 μL Lipofectamine 2000 reagent in 200 μL Opti-MEM. Forty-eight hours after transfection, the cells were treated with neomycin to select those with stable expression of the neomycin resistance gene. Thus, the stable cells were selected by detecting neomycin resistance after drug treatment (800 μg/mL) and single-cell clone isolation.

### 2.3. Virus Production and Transduction

Lentiviral vectors were produced by co-transfection of HEK293T cells with the psPAX2 envelope and pMD2.G packaging plasmids using the calcium phosphate transfection method (2 M CaCl2, 2 × HEPES buffered saline (pH 7.2)) [24]. Cells were plated in a 100 mm culture dish on the day before transfection. The cells were co-transfected with 5 µg of vector plasmids, 1.5 µg of psPAX2, and pMD2.G. The medium was changed 24 h after transfection, and the supernatant medium containing the virus was collected 48 and 72 h post-transfection. The collected medium was filtered through a 0.45 µm filter and mixed with the Lenti-X concentrator (Takara, Kyoto, Japan) in accordance with the manufacturer’s instructions. Viral pellets were stored in Opti-MEM and maintained at −70 °C. The cells in the 6-well tissue culture plates were infected with the medium containing virus and 8 mg/mL polybrene. Twenty-four hours after the infection, the medium was changed, and the cells were selected using puromycin (2 mg/mL).

### 2.4. Western Blotting

For the Western blot analysis, cells were collected in cell lysis buffer (150 mM NaCl, 50 mM HEPES [pH 7.5], and 1% NP40) containing protease inhibitor cocktail (Roche, Basel, Switzerland). Polypeptides in whole-cell lysates were resolved by SDS-PAGE and transferred onto immune-blot PVDF membrane filters (Bio-Rad, Hercules, CA, USA). Proteins were detected with a 1:1000 or 1:5000 dilution of primary antibodies using an ECL system (Dogen, Seoul, South Korea). Images were acquired using ChemiDoc-it 410 Imaging System (Analytik Jena, Upland, CA, USA) and ImageQuant LAS 4000 System (GE Healthcare, Waukesha, WI, USA). LNX1-specific antibodies were purchased from Lifespan Biosciences (Seattle, WA, USA). Cyclin D1- and cyclin E1-specific antibodies were purchased from Cell Signaling Technology (Beverly, MA, USA), and PARP-specific antibodies were purchased from GeneTex (San Antonio, TX, USA).

### 2.5. Reporter Plasmid Construction and Reporter Assay

The *LNX1* promoter region was amplified from HEK293 genomic DNA using the forward primer 5′-GGCTCGAGGGGATAGCCACACCTACCTTA-3′ and reverse primer 5′-GGAAGCTTGGGAAAGCATTGCTGAGACCT-3′. The amplified fragments were initially cloned into the pTOP TA V2 vector (Enzynomics, Daejeon, Korea, EZ011). The reporter plasmid construct was amplified and subsequently subcloned into the pGL3-promoter vector (Promega, Madison, WI, USA, E1641). For the reporter assay, cells were seeded in 24-well plates in DMEM for 18 h before transfection. Typically, cells in each well were transfected with less than 0.5 μg of the total DNA, and Renilla luciferase was used for normalization in each assay. To examine the cell signaling pathways, HEK293 and A549 cells were treated with cisplatin, topotecan, hydroxyurea, LPS, and PMA and irradiated with UV.

### 2.6. Quantitative RT-PCR

For quantitative RT-PCR, cells were harvested, and RNA was extracted using TRIzol (Thermo Fisher Scientific). Reverse transcription was conducted with an M-MLV RT kit (Enzynomics, Daejeon, South Korea) according to the manufacturer’s protocol, and PCR was performed using the StepOnePlus Real-Time PCR System (Thermo Fisher Scientific). *LNX1* mRNA was amplified using the forward primer 5′-TGAGCCCGGAGGAGTCATAA-3′ and reverse primer 5′-ATTCCAGCCACATGACCCAG-3′. For amplifying *p21* mRNA, the following pair of primers was used: forward 5′-CATGTGGACCTGTCACTGTCTTGTA-3′ and reverse 5′-GAAGATCAGCCGGCGTTTG-3′.

### 2.7. Cell-Cycle Analysis

A double-thymidine block was used to arrest A549 cells at the G1/S transition. A549 cells were synchronized by the addition of 2 mM thymidine from a 100 mM stock solution for 18 h. Next, the cells were washed with phosphate buffered saline (PBS), followed by the addition of regular growth medium (DMED supplemented with 10% FBS). Eight hours after the incubation, the medium was changed to that containing 2 mM thymidine and incubated for another 18 h. The cells were washed twice with PBS and replaced with fresh growth medium. The cells were collected at the indicated time points for the preparation of whole cell extracts. For cell-cycle analysis, the cells were washed and fixed with 70% ethanol. After centrifugation, the cells were resuspended in PBS containing 0.25 mg/mL propidium iodide (PI) and 10 mg/mL RNase A (Sigma, St. Louis, MO, USA). Next, the cell-cycle progression was analyzed using a FACSCalibur flow cytometer (Becton Dickinson, Mountain View, CA, USA).

### 2.8. Statistical Analysis

The results of the luciferase assay, Western blot, quantitative RT-PCR, cell viability assay, and cell-cycle distribution assay were evaluated using a 2-tailed Student’s *t*-test using Microsoft Excel 2016 software (Microsoft, Redmond, WA, USA). Statistical significance was set at *p* < 0.05.

## 3. Results

### 3.1. DNA Damage Decreases the Expression Levels of LNX1

To investigate the potential regulation of *LNX1* expression, we analyzed the nucleotide sequence of the *LNX1* promoter using the Ensemble database [25] and constructed reporter plasmids containing 1.5 kb promoter fragments relative to the transcription start site (TSS) of *LNX1*. Anticancer drugs such as cisplatin and topotecan, UV, and hydroxyurea induce DNA damage [26]; hence, we transfected A549 cells with reporter plasmids containing the *LNX1* promoter regions and treated the cells with DNA damage-inducing reagents. Although the cells with the control reporter (pGL3-promoter) did not exhibit any response to stimuli, a substantial reduction was observed with the reporter constructs containing *LNX1* promoter regions (Figure 1A). We tested whether each stimulus could regulate the expression levels of *LNX1* and found that the mRNA expression levels of *LNX1* decreased in a dose-dependent manner in response to each stimulus (Figure 1B and Appendix A). Additionally, we found the protein expression level of LNX1 was decreased by DNA damage (Figure 1C,D). These results suggest that DNA damage decreases the expression of *LNX1* at both mRNA and protein levels.

### 3.2. LPS, PMA, and Serum Addition Increase the Expression Levels of LNX1

Lipopolysaccharide (LPS) treatment leads to the early activation of the *NF-κB*, *IRF3*, and *MAPK* kinase signaling pathways [27,28,29]. Moreover, PMA is involved in the activation of the *PKC*-mediated *NF-κB* and *MAPK* signaling pathways [30,31]. Therefore, we examined whether LPS and PMA could regulate the expression of *LNX1*. Both LPS and PMA increased the promoter activity of *LNX1* (Figure 2A). Growth medium containing FBS provides many factors that are essential for cell growth. *LNX1* expression levels in cells grown in serum-free medium were lower than those in cells grown in growth medium. However, replacing the growth medium reactivated the expression of *LNX1* (Figure 2B and Appendix A). Furthermore, these factors increased the protein expression levels of LNX1 (Figure 2C,D). Therefore, signaling pathways associated with cell growth influence the expression of *LNX1*.

In addition, we conducted a bioinformatic analysis (AliBaba2.1, [32]) for potential binding sites for oncogenic transcription factors. The LNX1 promoter region (−500 bp) contains several oncogenic transcription factors, including NF-kB, c-myc and AP-1 (Appendix A). We also analyzed different publicly available gene expression datasets (GSE13333, GSE17511, GSE37219, and GSE6077) (Appendix A). These data included gene expression profiles in mouse serum response factor (SRF)-deficient mice (GSE13333) [33], the back skin from transgenic mice with the inhibitor of *NF-κB* kinase beta (IKK-β) overexpression (GSE17511) [34], osteoclasts lacking the transcription factor *NFATc1* (GSE37219) [35], and the lungs of transgenic embryos overexpressing *N-myc* in the lung epithelia (GSE6077) [36]. *LNX1* expression levels significantly decreased in SRF-deficient cells, *NFATc1*-deficient cells, and IKK-β-overexpressing cells. The expression levels of *LNX1* increased in proto-oncogene *N-myc* transgenic models. These data suggest that *LNX1* mRNA expression levels are affected by the growth signaling pathways containing AP-1, *NFATc1*, *N-myc*, and *NF-kB*.

### 3.3. LNX1 Expression Correlates with Cell-Cycle Progression

Since the expression levels of *LNX1* increased in growth medium, we examined whether this was related to cell-cycle progression. First, A549 cells were synchronized by using a double-thymidine block, which obstructs DNA replication and synchronizes cells at the G1-S border [37,38]. According to the DNA content of cells determined by PI, the cells progressed into S phase 4 h after the release of the block; after 6 h, the cells reached the G2 phase. (Figure 3A). We measured the mRNA expression levels of *LNX1* under the same conditions. *LNX1* expression levels decreased in G1/S arrest (0 h); however, these levels increased during cell-cycle progression (2–6 h) (Figure 3C). *LNX1* expression levels were correlated with the S/G2/M population of the cells (Figure 3D). These data suggest a relationship between *LNX1* and cell-cycle dynamics.

### 3.4. LNX1 KO Delays Cell-Cycle Progression

To examine the potential role of *LNX1* in the cell cycle, we generated *LNX1* KO A549 cells using the CRISPR-CRISPR associated protein 9 (CRISPR-Cas9) genome-editing system. Western blot analysis was conducted using anti-LNX1 antibodies and revealed that LNX1 expression levels decreased in A549 cells (Figure 4A). The expression levels of genes encoding cyclin D1 and cyclin E1 decreased in *LNX1* KO cells; however, those encoding the cyclin-dependent kinase inhibitors p16INK4A and p21WAF1/Cip1 increased (Figure 4A). These results suggest that *LNX1* KO cells exhibit aberrant cell-cycle dynamics.

To investigate whether *LNX1* KO contributes to cell-cycle progression, we examined the cell-cycle profiles of WT and *LNX1* KO cells. The WT and *LNX1* KO cells were synchronized by using a double-thymidine block, and the cell-cycle progression was determined by flow cytometry analysis (Figure 4B). The cell-cycle profiles were similar in WT and *LNX1* KO cells during G1/S arrest (0 h). However, the entry into the G2/M-phase (6 h after release) was delayed in *LNX1* KO cells. In WT cells, the progression through the G2/M-phase (47%) was completed after 6 h, and a significant fraction of *LNX1* KO1 cells (34%) and KO2 cells (27%) did not reach the G_2_/M-phase even after 6 h of release (Figure 4B). Consistent with this data, genes encoding cyclin D1 and cyclin E1 were found to be differentially expressed in *LNX1* KO cells compared with those in WT cells. In particular, expression levels of gene encoding cyclin E1 decreased and cell-cycle progression was delayed in *LNX1* KO cells at the release time points (Figure 4C,D). In addition, increased expression levels of the CDK inhibitor *p21* were observed in *LNX1* KO cells (Figure 4E).

### 3.5. LNX1 Overexpression Leads to Cell-Cycle Progression and Facilitates Cisplatin Resistance

Once we had found that the cell growth conditions increase expression of *LNX1* and disruption of *LNX1* delays cell-cycle progression, we wondered what effect increased expression of *LNX1* would have on the cell cycle. We found that the expression of genes encoding cyclin D1 and cyclin E1 was induced in cells with an increased dosage of transfected plasmid encoding *LNX1* (Figure 5A–C). The endogenous expression levels of genes encoding cyclin D1 and cyclin E1 increased depending on the amount of transfected plasmid encoding *LNX1*, thus indicating that the upregulation of these genes was dependent on the dose of *LNX1*. We constructed HEK293 cell lines that stably expressed *LNX1* for subsequent analysis (Figure 5D). *LNX1*-expressing HEK293 cells were then incubated with thymidine to synchronize cells and were harvested at the indicated time points after the release of thymidine. The overexpression of *LNX1* increased the proportion of cycling cells (S/G2/M phase) (Figure 5E,F). In addition, the proliferation of cells stably expressing *LNX1* increased, in comparison with that of the control cells (Figure 5G). These data suggest that overexpression of *LNX1* leads to cell-cycle progression.

As *LNX1* expression induced the cell cycle, we sought to determine its role in cancer therapy using cisplatin. We treated the control and *LNX1*-expressing cells with cisplatin and measured cell viability using the MTT assay. Trypan blue-stained cells were counted using the cell counter. We found that the expression of *LNX1* reduced cisplatin-mediated cell death (Figure 6A–C). Transient *LNX1* overexpression resisted cisplatin-induced apoptosis (Figure 6A). In addition, *LNX1*-overexpressing cells showed decreased cell death following cisplatin treatment (Figure 6B,C). Further, in cells stably expressing *LNX1*, the subG1 population and aberrant distribution of cell cycles induced by cisplatin were reduced (Figure 6D). Additionally, upon cisplatin treatment, the cleavage of PARP1 (an apoptosis marker protein) decreased in cells stably expressing *LNX1* compared with that in control cells (Figure 6E).

As the overexpression of *LNX1* decreased cisplatin-induced cell death in HEK293 cells, we examined whether *LNX1*-overexpressing cancer cells were resistant to cisplatin treatment. *LNX1*-overexpressing cells were generated with A549, MCF7 and H460 cells and the expression of cyclin D1 and cyclin E1 were examined. We found cyclin D1 expression is increased in *LNX1*-overexpressing cells (Figure 7A). Next, we treated control and *LNX1*-overexpressing cells with cisplatin, and found that *LNX1* expression resulted in enhanced cell viability upon cisplatin treatment (Figure 7B). These data suggest that the overexpression of *LNX1* contributes to cisplatin resistance.

## 4. Discussion

In a previous study, we reported that *LNX1* acts as an oncogene and found that the expression levels of *LNX1* increased in cancers using the TCGA database [16]. However, the factors that regulate *LNX1* expression in cells was unknown. In this study, we found that several stimuli regulate *LNX1* expression (Figure 1 and Figure 2). DNA damaging agents, including the anticancer drugs cisplatin and topotecan, UV, and hydroxyurea, decreased the expression levels of *LNX1*. Moreover, the promoter activity and mRNA and protein levels of *LNX1* were decreased by these stimuli. We analyzed the promoter region of *LNX1* based on the Ensemble database [25] and constructed a reporter plasmid containing the *LNX1* promoter region (−1.5 kb upstream from TSS).

LPS, PMA, and replacement of serum-free medium with FBS-containing medium increased *LNX1* expression levels, including promoter activity and levels of mRNA and protein (Figure 2). LPS leads to the early activation of the *NF-κB*, *IRF3*, and *MAPK* kinase signaling pathways [27,28,29]. Additionally, PMA is involved in the *PKC*-mediated *NF-κB* and *MAPK* signaling pathways [30,31]. Therefore, we expected that signaling associated with the regulation of cell growth, such as *MAPK* and *NF-kB*, was involved in the regulation of *LNX1* expression. Other studies have shown that cell-growth-associated signaling pathways, including *N-myc*, *NF-Kb*, and *NFATc1*, regulate the expression of *LNX1*. Using publicly available gene expression datasets (GSE13333, GSE17511, GSE37219, GSE6077), several studies have implicated that the mRNA expression of *LNX1* is affected by the growth signaling pathway containing *NFATc1*, *N-myc*, and *NF-kB* [33,34,35,36]. In particular, *LNX1* expression significantly decreased in SRF-deficient mice. In our results, the expression of *LNX1* decreased in cells grown in serum-free medium compared with that in cells grown in growth medium. Replacement of the growth medium with serum-free medium induced the expression of *LNX1*. These data suggest that the expression of *LNX1* is associated with oncogenic responses, such as growth factors and other oncogenes.

As we anticipated that cell growth could be affected by *LNX1*, we analyzed its relationship with the cell cycle. We applied the widely used double-thymidine-block method to analyze the cell cycle [38,39]. The double-thymidine block causes G1/S arrest, and replacement with fresh medium aids progression to the next stage of the cell cycle. The expression levels of *LNX1* decreased in synchronous cells compared with those in asynchronous cells. However, *LNX1* expression levels increased after thymidine release. Therefore, we found a correlation between *LNX1* expression and S/G2/M populations (Figure 3). When we used hydroxyurea to block the G1 to S transition, we observed that the expression pattern of *LNX1* and its correlation with the cell-cycle distribution were similar to those when thymidine was used. Moreover, CRISPR-Cas9-mediated *LNX1* KO cells delayed the progression of the cell cycle. The population of G2/M cells after 6 h of thymidine release was decreased in *LNX1* KO (Figure 4B). This result is consistent with the previous report that downregulation of LNX1 by siRNA induces cell-cycle arrest in HEK293 cells [40]. The expression levels of genes encoding cyclin D1 and cyclin E1 decreased in *LNX1* KO cells, and the cyclin kinetics collapsed in the thymidine-mediated cell-cycle analysis. In particular, the expression level of gene-encoding cyclin E1 decreased and cell-cycle progression was consequently delayed in *LNX1* KO cells (Figure 4D). This could be because gene encoding cyclin E1 is a crucial regulator and is dynamically expressed during the G1 to S transition [41,42]. In *LNX1* overexpression experiments, the expression level of the gene-encoding cyclin D1 was found to be higher than that of gene-encoding cyclin E1 (Figure 5A). As most cells remained in the G1 phase, gene-encoding cyclin D1 was more dominant than that encoding cyclin E1 in this stage of the cell cycle [43,44].

In a previous study, we demonstrated that *LNX1* acts as a proto-oncogene by inhibiting *p53* stability [16]. *p53* plays a crucial role in DNA repair and is a determinant of the development of cisplatin resistance [45]. The main mechanism underlying the anticancer property of cisplatin is DNA damage caused by binding platinum to DNA, thereby forming DNA adducts. The DNA damage by cisplatin induces cell-cycle arrest and initiates apoptosis in fast-cycling cells [21,46]. However, cancer stem cells remain in a quiescent, slow-proliferating state and escape from chemotherapy, including cisplatin treatment that targets fast-cycling cells [46].

In this report, we studied the regulation of *LNX1* expression and its consequences. Cisplatin treatment resulted in the downregulation of *LNX1*, and *LNX1* overexpression contributes to enhanced cell viability upon cisplatin treatment. These results support the notion that *LNX1* expression regulates cisplatin resistance or cisplatin sensitivity. We found that *LNX1* contributes to cell-cycle progression and the stability of *p53* [16]; therefore, we examined whether it is related to cisplatin resistance or sensitivity. The overexpression of *LNX1* reduced cell death following cisplatin treatment. The subG1 population in cisplatin-treated WT cells was higher than that in *LNX1*-stable cells, and the extent of the collapse of the cell-cycle distribution of WT cells was greater than that of cisplatin-treated *LNX1*-stable cells (Figure 5). Although *LNX1* overexpression accelerated the cell cycle, the sensitivity of cisplatin did not increase in *LNX1*-expressing cells. It is possible that cisplatin has various mechanisms that induce cell death and compensate for the absence of *LNX1* [47,48,49]. Moreover, apoptosis signaling involving *p53* is directly associated with cell death induced by cisplatin. Knocking out *LNX1* did not influence the viability of cisplatin. Both WT and *LNX1* KO cells might have efficient *p53* activity that can induce cell death by cisplatin treatment. However, because the overexpression of *LNX1* can lead to the degradation of *p53*, cell death is reduced by cisplatin treatment [16,50]. Further studies are required to elucidate the relationship between *p53* and *LNX1* expression in cisplatin resistance and the role of *LNX1* in DNA damage response. In addition, we need to confirm the role of *LNX1* in various other cancer cell lines and in vivo.

## 5. Conclusions

Here, we demonstrate that *LNX1* expression is decreased by DNA damage, and increased by cell cycle progression. We also showed that *LNX1* knockout results in a delay in cell cycle progression and *LNX1* overexpression activates cell cycle progression. Finally, *LNX1* overexpression increased resistance to cisplatin treatment. These results indicate that elevated level of *LNX1* in cancer cells contributes to tumor growth and enhanced drug resistance.

## Figures and Tables

**Figure 1 cancers-13-04066-f001:**
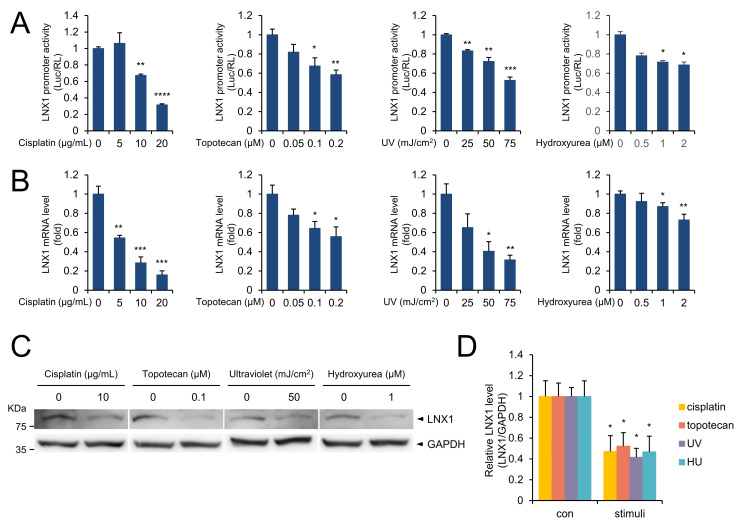
DNA damage decreased the *LNX1* expression levels. (**A**) A549 cells were transfected with pGL3 plasmids containing *LNX1* promoter sequence. After 24 h, cells were treated with the indicated drugs for 24 h. Cells were irradiated with UV and incubated for 4 h, and luciferase activity was measured. Relative luciferase activity was normalized to Renilla luciferase activity and was represented as a fold change compared with control. Experiments were performed in triplicate, and standard deviation is shown as control vs. drug-treated cells, *: *p* < 0.05, **: *p* < 0.01, ***: *p* < 0.005, and ****: *p* < 0.001. (**B**) The indicated stimuli decreased *LNX1* mRNA levels. Quantitative RT-PCR was used to measure the mRNA expression levels of *LNX1*. Control vs. drug-treated cells, *: *p* < 0.05, **: *p* < 0.01, and ***: *p* < 0.005. (**C**,**D**) *LNX1* protein levels decreased in response to the indicated stimuli. Each cell lysate was probed with the indicated antibodies. The uncropped Western Blot images can be found in Appendix A.

**Figure 2 cancers-13-04066-f002:**
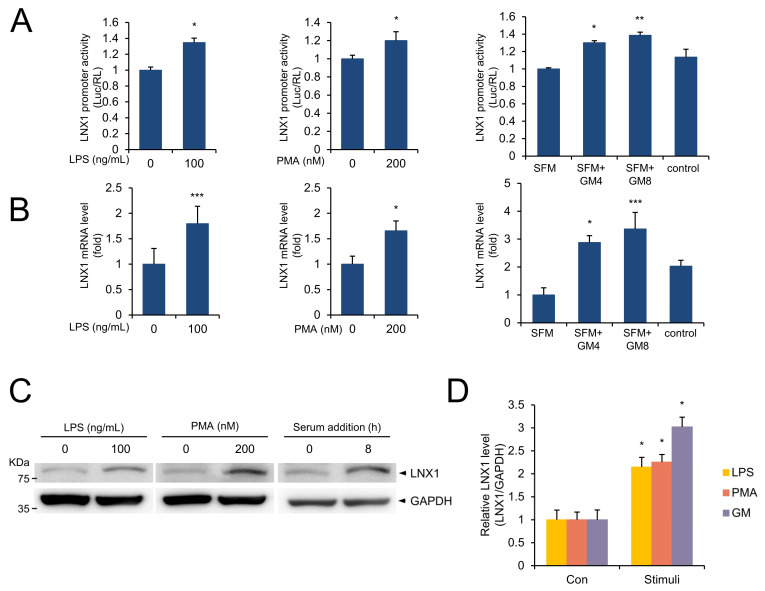
LPS, PMA, and serum addition increase the *LNX1* expression levels. (**A**) A549 cells were transfected with pGL3-LNX1 plasmids. After 24 h, cells were incubated with the indicated concentrations of LPS, PMA or serum-free medium. Serum-free medium was replaced with growth medium (GM) and cells were incubated for the indicated hours. Relative luciferase activity was measured and represented as a fold change compared with control. Experiments were performed in triplicate, and standard deviation is shown as control vs. drug-treated cells, *: *p* < 0.05, **: *p* < 0.01, ***: *p* < 0.005 (**B**) The indicated stimuli increased *LNX1* mRNA levels. Expression levels were calculated by quantitative real-time PCR analysis. The results are represented as relative mRNA levels. (**C**,**D**) *LNX1* protein levels increased in cells grown in LPS, PMA, and FBS-containing medium. Each cell lysate was probed with the indicated antibodies and quantified. The uncropped Western Blot images can be found in Appendix A.

**Figure 3 cancers-13-04066-f003:**
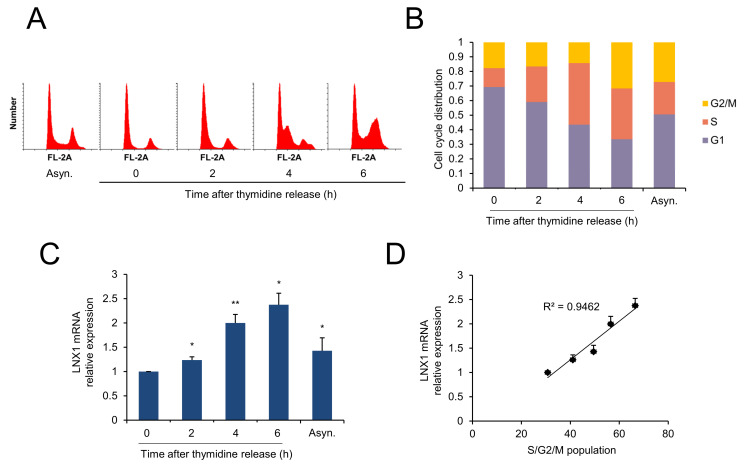
*LNX1* expression correlates with cell-cycle progression. A549 cells were synchronized using a double-thymidine block. Once released, cells were collected at the indicated time points. (**A**,**B**) Cell-cycle distribution was obtained by flow cytometry analysis of the DNA of PI-stained cells. (**C**) Expression levels were calculated by quantitative real-time PCR analysis. Synchronous vs. thymidine release or asynchronous, *: *p* < 0.05 and **: *p* < 0.01, (**D**) *LNX1* expression is correlated with S, G2, and M populations. The R2 values were calculated and have been shown in the graphs.

**Figure 4 cancers-13-04066-f004:**
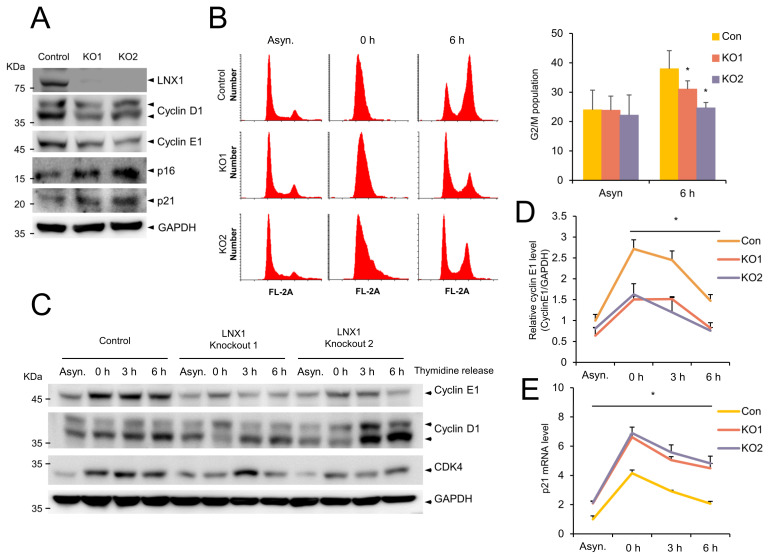
Knocking out *LNX1* delays cell-cycle progression. (**A**) Expression levels of genes encoding the cell-cycle markers cyclin D1 and cyclin E1 decreased in *LNX1* knockout (KO) cells. *LNX1* KO cell lines were generated using the lentiviral-mediated CRISPR-Cas9 genome editing. A549 cells were infected with the lentivirus encoding *LNX1* gRNA. Equal amounts of cell lysates were probed with the indicated antibodies. (**B**) G2 and M populations decreased in *LNX1* KO cells. Cells were synchronized by a double-thymidine block. Once released, cells were collected at indicated time points. Cell-cycle distributions were obtained following flow cytometry analysis of DNA in PI-stained cells. Control vs. *LNX1* KO, *: *p* < 0.05 (*n* = 3). (**C**) Knocking out *LNX1* disrupts the cell-cycle dynamics and expression of genes encoding cyclin D1 and cyclin E1. Control and *LNX1* KO cells were synchronized by a double thymidine block. Once released, cell lysates were collected at indicated time points. Equal amounts of cell lysates were probed with the indicated antibodies. (**D**) The bands of gene encoding cyclin E1 were quantified, and the relative expression levels are shown in the graph. Control cells vs. *LNX1* KO cells, *: *p* < 0.05. (**E**) mRNA levels of *p21* were measured based on quantitative real-time PCR analysis. Control cells vs. *LNX1* KO cells, *: *p* < 0.05. The uncropped Western Blot images can be found in Appendix A.

**Figure 5 cancers-13-04066-f005:**
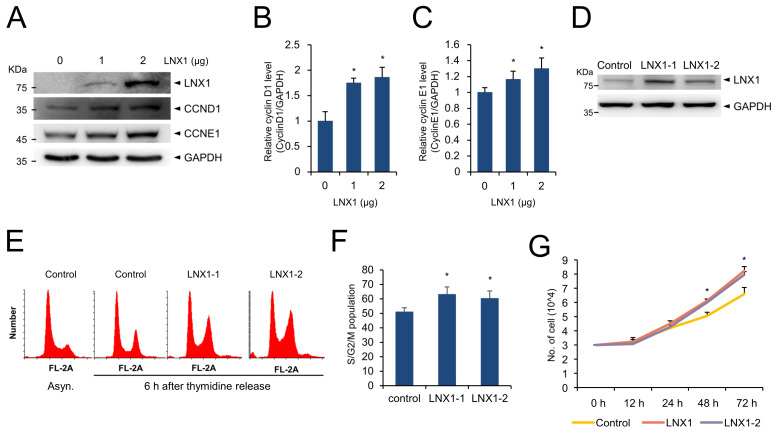
Overexpression of *LNX1* promotes cell-cycle progression. (**A**) The expression levels of genes encoding cyclin D1 and cyclin E1 increased in a dose-dependent manner in HEK293 cells that were transfected with plasmids encoding *LNX1*. Cell lysates were probed with indicated antibodies. (**B**,**C**) The bands of genes encoding cyclin D1 and cyclin E1 were quantified, and the relative expression levels are shown in the graph. Control cells vs. *LNX1*-overexpressing cells, *: *p* < 0.05. (**D**) HEK293 cells with stable expression of *LNX1* were developed, and (**E**,**F**) cell-cycle distribution was analyzed. Cells were synchronized by using double-thymidine block and harvested at 6 h after thymidine release. The cell-cycle profiles were obtained following flow cytometry analysis of DNA in PI-stained cells. (**G**) Equal numbers of control and *LNX1* (stable expression) cells were seeded and incubated for the indicated hours. Each cell was counted via Vi-Cell XR counter (Beckman Coulter, Brea, CA, USA), which is a trypan blue-based automatic cell counter. The uncropped Western Blot images can be found in Appendix A.

**Figure 6 cancers-13-04066-f006:**
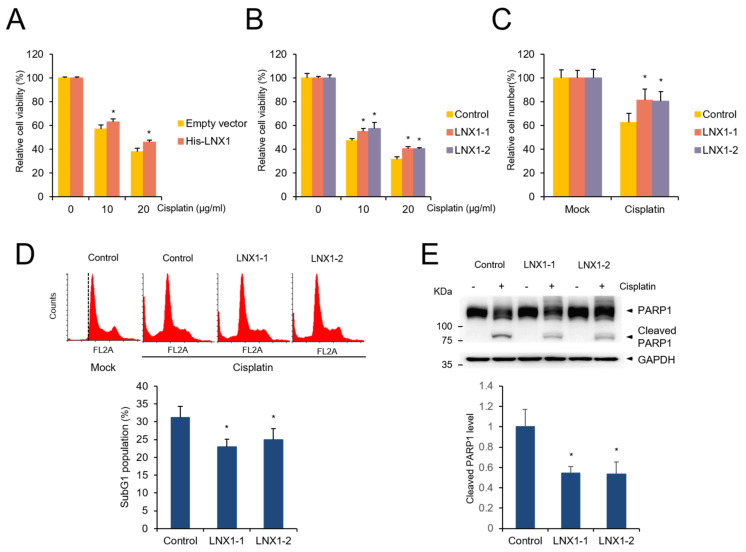
Overexpression of *LNX1* increases resistance to cisplatin. (**A**) HEK293 cells were transfected with plasmids encoding *LNX1* and treated with the indicated concentrations of cisplatin for 24 h. (**B**) Control and *LNX1* (stable expression) cells were treated with cisplatin. Cell viability was measured using MTT assay. Control cells vs. *LNX1*-overexpressing cells or cells stably expressing *LNX1*, *: *p* < 0.05 (**C**) HEK293 cells stably expressing *LNX1* were treated with 10 µg/mL of cisplatin for 24 h. Live cells were counted using Vi-Cell XR counter, which is a trypan blue-based automatic cell counter. (**D**) Upon cisplatin treatment, subG1 population was reduced in cells stably expressing *LNX1*. Cells that were treated with 10 µg/mL of cisplatin for 24 h were stained with PI, and the subG1 population was analyzed using flow cytometry analysis. Control cells vs. cells stably expressing *LNX1*, *: *p* < 0.05. (**E**) Cisplatin-treated control and cells stably expressing *LNX1* were harvested 24 h after treatment. Cell lysates were probed with PARP1 antibodies. Control cells vs. cells stably expressing *LNX1*, *: *p* < 0.05. The uncropped Western Blot images can be found in Appendix A.

**Figure 7 cancers-13-04066-f007:**
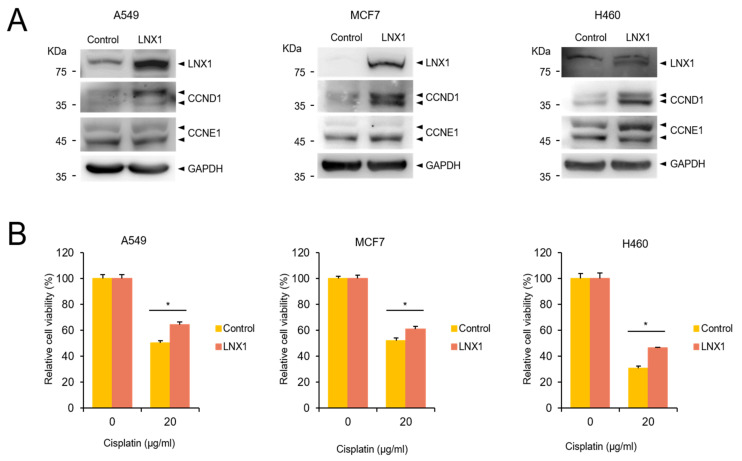
Overexpression of *LNX1* increases resistance to cisplatin in cancer cells. (**A**) *LNX1*-overexpression cells were generated using A549, MCF7 and H460 cells, and the cell lysates were subject to Western blots with indicated antibodies. (**B**) Control and *LNX1* (stable expression) cells were treated with cisplatin. Cell viability was measured using MTT assay. Control cells vs. *LNX1*-overexpressing cells or cells stably expressing *LNX1*, *: *p* < 0.05. The uncropped Western Blot images can be found in Appendix A.

## Data Availability

Not applicable.

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
