# Peer review of "LNX1 Contributes to Cell Cycle Progression and Cisplatin Resistance"

_cancers, 2021, doi:10.3390/cancers13164066_

Round 1
Reviewer 1 Report
The authors have addressed the issues raised in my review. Form my perspective the paper can now be published.
One small correction: The colours in the key and on the graph in the left panel of figure S2B do not match. This should be corrected.
Reviewer 2 Report
The authors have addressed all the points raised previously. I have nothing more to add.
This manuscript is a resubmission of an earlier submission. The following is a list of the peer review reports and author responses from that submission.
Round 1
Reviewer 1 Report
LNX1 is a ring-domain E3 ligase and is found to be overexpressed in several cancers. The authors previously reported that LNX1 down-regulates tumor suppressor p53, which in turn exerts its oncogenic properties. In this manuscript, the authors showed that LNX1 could be down-regulated by several DNA damage agents, including cisplatin, and up-regulated by factors promoting cell growth. They also manipulated the level of LNX1 in cells and showed that LNX1 promotes cell cycle progression and cisplatin resistance. Overall, the manuscript is well-written, and the data are well-presented to support the conclusions. However, several significant caveats were identified. First, the manuscript lacks profound mechanistic studies. Does cisplatin-mediated down-regulation of LNX1 suggest that the level of LNX1 could be an indicator for cisplatin sensitivity? Second, Figures1-2 and 4 only used a single lung cancer cell line A549. Multiple cancer cell lines are needed to support the conclusions. The cell line being used for Figure 3 is not clearly stated. For Figures 5 and 6, the HEK293T cell line, a non-transformed cell line, was used. Again, here cancer cell lines should be utilized. The in vivo xenograft mouse studies using control vs. LNX1 knockdown cell lines treated with vehicle or cisplatin should be performed.
Reviewer 2 Report
LNX1 contributes to cell cycle progression and cisplatin resistance
Authors in the manuscript have identified the oncogenic role of LNX1 in cancer cells. DNA damage due to platinum therapy or UV leads to increased LXN1 expression in cancer cells. Exogenous over-expression of LXN1 in turn leads to cell cycle progression in cancer cells. Treatment of cancer cells with LPS or PMA induces increased expression of LXN1 in cancer cells. Further knockdown of LXN1 expression in cancer cells leads to delayed progression of cell cycle by increasing the expression of p16 and p21. Expression of LXN1 also increases as the cancer cell progresses from one cell cycle stage to the other. Finally LXN1 expression directly correlates with cisplatin resistance in cancer cells.
Pros
-
The study is well written and the results justify the conclusion drawn.
-
LXN1 being a therapeutic target for cancer cell treatment provides novelty aspect to this manuscript.
Cons
-
In Figure 1C topotecan concentration mentioned over western blot is mentioned as 0.1µM whereas the mRNA studies before in Fig. 1B that topotecan concentration 0.5µM doesnt significantly alter mRNA levels. I suppose it is a typographical error.
-
Line 200, figure 2A represents the promoter activity of LXN1 and not the mRNA levels of LXN1 as claimed by the authors. In line 203, Figure 2B denotes the mRNA expression of LXN1. Serum free media studies are different from 2A and 2B and as such should be denoted separately from Figure 2A and 2B.
-
Line 231, claims 6h post thymidine block most of the cells reached G2 phase but the cell cycle distribution shows only 30% of the cells in G2 phase.
-
Figure 3A, 4B, 5E and 6D requires Y-axis and X-axis label that determine which channels have been used for flow cytometry and the data analysis.
Reviewer 3 Report
The manuscript provides some novel information on how the LNX1 expression at the transcriptional level – an area that has not been studied to any great extent to date. It also provides some insights into the involvement of LNX1 in the regulation of cell cycle and a potential role in resistance to chemotherapeutic agents. The data are generally very clear and well presented. The manuscript is well written and logical in its progression.
Nevertheless, the following changes would improve the manuscript.
1.
The relatively short format of the manuscript limits the literature related to LNX proteins that can be cited in the introduction and discussion. However, there is some additional relevant literature that should be mentioned. In particular the results of the present study should be discussed in comparison to the findings of the following article on a very related topic.
Zheng, D., Gu, S., Li, Y., Ji, C., Xie, Y. and Mao, Y. (2011) A global genomic view on LNX siRNA-mediated cell cycle arrest. Mol. Biol. Rep. 38, 2771–2783, https://doi.org/10.1007/s11033-010-0422-6
Also, the following sentence in the introduction “Furthermore, several studies associated with cancer and LNX1 have suggested its role as a tumor regulator in various cancers, including gliomas, hepatocellular carcinoma, and colorectal carcinoma [11-13].” gives the impression that LNX1 has been implicated in liver cancer. In fact, as far as I am aware, the Hu et al paper cited here (reference #13) relates to PDZRN4/LNX4 and not LNX1. Since LNX1 is more homologous to LNX2, it might be more relevant to mention literature related to LNX2 and cancer such as:
Camps, J., Pitt, J.J., Emons, G., Hummon, A.B., Case, C.M., Grade, M. et al. (2013) Genetic Amplification of the NOTCH modulator LNX2 upregulates the WNT/beta-catenin pathway in colorectal cancer. Cancer Res. 73, 2003–2013, https://doi.org/10.1158/0008-5472.CAN-12-3159
In addition, while there is not space to extensively discuss the literature on LNX proteins, it might be worth referring the reader to two recent review articles on LNX proteins, both of which have sections on LNX proteins and cancer.
Hong et al (2020) Molecules, 25, 5938; doi:10.3390/molecules25245938
Young (2018) Neuronal Signaling 2 NS20170191 https://doi.org/10.1042/NS20170191
2.
In figure 1 a 1.5 kb promoter fragment of the Lnx1 gene was used in reporter assays. It would be interesting to present an analysis of any transcription factor binding site present in this sequence, in particular any that might explain the regulation of Lnx1 expression by the stimuli used in Fig 1 and 2. This would pave the way for more mechanistic studies of Lnx1 transcriptional regulation.
3.
The data related to cisplatin resistance (Figure 6) employ HEK cells. What is the rationale for using HEK cells rather than a cancer-derived cell line to study this phenomenon? This should be explained. The biological and clinical relevance of these findings would be strengthened by replicating them a cisplatin resistant cancer cell line – if such is available.